Integrating physical and tactical factors in football using positional data: a systematic review

Teixeira José Eduardo 1 2 3
Forte Pedro 1 2 4
Ferraz Ricardo 1 5
http://orcid.org/0000-0001-9000-5419 Branquinho Luís 1 4
Silva António José 1 6
Monteiro António Miguel 1 2
http://orcid.org/0000-0001-7071-2116 Barbosa Tiago M. 1 2 barbosa@ipb.pt
1 Research Center in Sports Sciences, Health Sciences and Human Development , Vila Real , Portugal
2 Department of Sport Sciences and Physical Education, Instituto Politécnico de Bragança , Bragança , Portugal
3 Department of Sport Sciences, Polytechnic Institute of Guarda , Guarda , Portugal
4 Department of Sports, Higher Institute of Educational Sciences of the Douro , Penafiel , Portugal
5 Sport Sciences Department, University of Beira Interior , Covilhã , Portugal
6 University of Trás-os-Montes and Alto Douro , Vila Real , Portugal
Chen Yung-Sheng
Electronic publication date: 2022 Nov 14
Publication date: 2022
Volume: 10
Electronic Location ID: e14381
Received 2022 Jul 18; Accepted 2022 Oct 21
Copyright: © 2022 Teixeira et al.
Copyright year: 2022
Copyright holder: Teixeira et al.
License: This is an open access article distributed under the terms of the Creative Commons Attribution License, which permits unrestricted use, distribution, reproduction and adaptation in any medium and for any purpose provided that it is properly attributed. For attribution, the original author(s), title, publication source (PeerJ) and either DOI or URL of the article must be cited.
License URL: https://creativecommons.org/licenses/by/4.0/

Keywords: Tracking systems, Movement, Complexity, Training, Match, Sports, Soccer

Funding: Portuguese Foundation for Science and Technology UIDB/04045/2020 This research was supported by Portuguese Foundation for Science and Technology, I.P. (project UIDB/04045/2020). The funders had no role in study design, data collection and analysis, decision to publish, or preparation of the manuscript.

==============================
Background

Positional data have been used to capture physical and tactical factors in football, however current research is now looking to apply spatiotemporal parameters from an integrative perspective. Thus, the aim of this article was to systematically review the published articles that integrate physical and tactical variables in football using positional data.

Methods and Materials

Following the Preferred Reporting Item for Systematic Reviews and Meta-analyses (PRISMA), a systematic search of relevant English-language articles was performed from earliest record to August 2021. The methodological quality of the studies was evaluated using the modified Downs and Black Quality Index (observational and cross-sectional studies) and the Physiotherapy Evidence Database (PEDro) scale (intervention studies).

Results

The literature search returned 982 articles (WoS = 495; PubMed = 232 and SportDiscus = 255). After screening, 26 full-text articles met the inclusion criteria and data extraction was conducted. All studies considered the integration of physical and tactical variables in football using positional data (n = 26). Other dimensions were also reported, such as psychophysiological and technical factors, however the results of these approaches were not the focus of the analysis (n = 5). Quasi-experimental approaches considered training sets (n = 20) and match contexts (n = 6). One study analysed both training and play insights. Small sided-games (SSG) were the most common training task formats in the reviewed studies, with only three articles addressing medium-sided (MSG) (n = 1) and large-sided games (LSG) (n = 2), respectively.

Conclusions

Among the current systematic review, the physical data can be integrated by player’s movement speed. Positional datasets can be computed by spatial movement, complex indexes, playing areas, intra-team and inter-team dyads. Futures researches should consider applying positional data in women’s football environments and explore the representativeness of the MSG and LSG.

Introduction

Football can be characterized as a complex and dynamical system where the players collect ecological information to make decisions, allowing them to gather numerical and spatial advantage through the phases of play with a goal-orientation (Duarte et al., 2012a; Folgado et al., 2014a; Hewitt, Greenham & Norton, 2016). Hence, football players carry out intermittent movements to perform individual and collective tactical actions (Clemente et al., 2020; Duarte et al., 2012b; Low et al., 2019). Tracking systems have been used to compute spatiotemporal measures and assess players’ positions during training and match settings (Lames, Erdmann & Walter, 2010; Marcelino et al., 2020; Pol et al., 2020). Positional data can be captured at different frequencies by tracking systems, such as, global navigation satellite systems (GNSS) or global positioning systems (GPS) (Beato et al., 2018; Rago et al., 2020; Teixeira et al., 2021a), local radio-based local positioning (LPM) (Hoppe et al., 2018; Leser, Baca & Ogris, 2011; Ogris et al., 2012) and computerized-video or optical-based tracking systems (Beato & Jamil, 2018; Castellano, Alvarez-Pastor & Bradley, 2014; Di Salvo & Marco, 2006). The players and ball positioning can be computed by Cartesian and Euclidian coordinates (xx, yy) contextualizing the physical demands on the tactical behaviour (Carrilho et al., 2020; Clemente et al., 2013; Low et al., 2019; Memmert, Lemmink & Sampaio, 2017). However, some of the above mentioned tracking methods do not allow to gather information on the player-ball-goal position (Carrilho et al., 2020; Vidal-Codina et al., 2022), opponent-adaptive play strategy (Memmert, 2021; Ranjitha, Nathan & Joseph, 2020) and individual tactical behavior (Laakso et al., 2022; Reis & Almeida, 2020).

Furthermore, tracking systems generate a large amount and variety of data that can be used for performance analysis in football (Rein & Memmert, 2016; Rojas-Valverde et al., 2019). Notwithstanding, it is paramount implementing multidisciplinary frameworks underpinned by sports science and computer science, making use of big data methodology, new computational procedures to extract, process and analyse data that yield practical information with an impact on training and match performance (Rico-González et al., 2021). However, integrating players’ physical performance with match-related contextual factors and tactical behaviours continues to be a challenge in football science (Teixeira et al., 2022b). Moreover, the performance analysis in football needs a multidimensional approach to capture the adaptive individual and collective behaviour (Carling et al., 2014; Gonçalves et al., 2019). This multifactorial phenomenon depends on the interplay of physical, tactical and technical drivers (Bradley & Ade, 2018; Paul, Bradley & Nassis, 2015). Thus, performance analysis in football is now focused on applying the spatiotemporal parameters from an integrative perspective (Praça et al., 2022).

A growing number of reviews and meta-analyses have been published on this topic and focusing on training and match settings (Low et al., 2019; Rago et al., 2020; Teixeira et al., 2021a). Nevertheless, previous reviews have considered each performance factor independently, describing tactical behaviour independently from physical demands (Coito et al., 2022; Ometto et al., 2018). Therefore, it is important to understand the main methodological procedures to conduct an integrative analysis of physical and tactical performance in training and match in football. Also, the published studies have calculated different physical and tactical measures by tracking positional data, wherefore a procedural standardization is needed to progress towards integrative approaches (Teixeira et al., 2022a). Thus, the aim of this study was to systematically review the published articles that integrate physical and tactical variables in football using positional data.

Methods

Literature search strategy

The literature search strategy was registered on the International Platform of Registered Systematic Review and Meta-Analysis Protocols with the number 202270030 (DOI 10.37766/inplasy2022.7.0030). The protocol was designed in accordance with ‘The Preferred Reporting Items for Systematic Reviews and Meta-Analyses’ (PRISMA) guidelines and the ‘Population-Intervention-Comparators-Outcomes’ (PICOS) (Page et al., 2021). The literature search was conducted on three databases: PubMed/Medline, Web of Science (WoS, including all Web of Science Core Collection: Citation Indexes), and SportDiscus. The studies were searched using a Boolean string with specific keywords (Table 1).

Table 1 Search terms and following keywords in the screening procedures of systematic review.

Search term		Keywords	
Population	1	“soccer” OR “football” OR “Association football”	
Intervention	2	“integrated” OR “integration” OR “comparison” OR “integration”	
Comparison/
outcomes	3	Physiological set: “training load” OR “external training load” OR “internal training load” OR “physical performance” OR “physiological performance” OR “physical response” OR “physical demands” OR “physiological response” OR “physiological demands” OR “activity profile” OR “time-motion” OR “workload” OR “work-rate” OR “loading” OR “match running performance” OR “match load” OR “match demands” OR “weekly load” OR “heart rate” OR “TRIMP” OR “perceived exertion” OR “distances” OR “sprint” OR “acceleration” OR “deceleration” OR “metabolic power” OR “energy cost” OR “high intensity” OR “running” OR “conditioning” OR “fitness” OR “biomechanics” OR “kinetic” OR “kinematic” OR “physiology”	
Positional data: “positional” OR “positioning” OR “behavioral data” OR “behaviour data” OR “tactical behavior” OR “tactical behaviour” OR “collective behavior” OR “collective behaviour” OR “team behavior” OR “team behavior” OR “movement behavior” OR “movement behaviour” OR “patterns” OR “constraints” OR “interpersonal coordination” OR “inter-personal coordination” OR “intra-team dyads” OR “inter-team dyads” OR “synchronization” OR “synergy” OR “tactical adjustments” OR “game dynamics” OR “dynamic” OR “variability” OR “stability” OR “regularity” OR “predictability” OR “spatial-temporal” OR “spatio-temporal” OR “complex systems” OR “dynamical systems” OR “complexity” OR “self-organization” OR “self-similarity” OR “self-organization” OR “chaos”	
Boolean phrase	4	(((#4) AND #3) AND #2) AND #1	

The literature search was performed between April and May 2022 by an independently author (J.E.T) and checked by a second author (P.F.). Discrepancies between authors in the study selection were solved by a third reviewer (T.M.B). Double-check review is recommended in PRISMA guidelines (Page et al., 2021). The literature search was limited to peer-reviewed articles and authors did not prioritize authors or journals.

Selection criteria

The selection criteria followed PICOS approach: (1) Population: amateur, semi-professional and professional football players (aged ≥10 years); (2) Intervention: integration of physical and tactical measures using spatiotemporal datasets; (3) Comparison: physical and tactical variables; (4) Outcomes: tracking, positional and time-series data; (5) Study design: original experimental and quasi-experimental trials (e.g., randomized controlled trial, cohort studies or cross-sectional studies).

On this basis, the inclusion criteria used for article selection were: (1) original article focused on adult and youth football players of both sexes; (2) studies with screening procedures based on physical and tactical measures using tracking and positional data; (3) studies that used spatiotemporal parameters to assess physical data; (4) studies that used positional and tracking data to measures spatiotemporal and tactical variables through time-series; (3) other performance factors as psychophysiological, technical and contextual factors were not excluded from the present review if both variables of interest (i.e., physical and tactical measures) were part of the experimental design; (5) studies of human physical performance in the field of sport science; (6) original articles published in peer-review journals; (7) full text available in English; (8) reported sample and screening procedures (e.g., data collection, study design, instruments, and the outcomes).

Otherwise, exclusion criteria were: (1) original articles about positional data in individual sports, team sports, and other football codes (e.g., Australian Football, Gaelic Football, Union and/or Seven Rugby); (2) studies that analysed none or only one of the performance drivers (i.e., only physical or tactical measures); (3) studies which integrate several performance factors, but did not combine the two domains of interest, even if one single driver is integrated; (4) studies that measured physical outcomes by field-based or laboratory tests rather than tracking and positional data; (5) studies that reported tactical variables collected by notational analysis or other methodological procedures that did not assess spatiotemporal time-series; (6) others research fields and non-human participants; (7) articles with poor quality in the description of study sample and screening procedures (e.g., data collection, study design, instruments, and the measures) according to PEDro and Downs and Black scales; (8) reviews, conference abstract/papers, surveys, opinion pieces, commentaries, books, periodicals, editorials, case studies, non-peer-reviewed articles, masters dissertations and doctoral theses.

Quality assessment

Methods quality was assessed by the modified Downs and Black Quality Index (cross-sectional studies) and the Physiotherapy Evidence Database (PEDro) scale (intervention studies) as done in previous systematics reviews (Downs & Black, 1998; Maher et al., 2003). For cross-sectional studies, the modified Downs and Black Index was used and is a 14-item scale, with larger scores deemed of studies with better quality. For intervention studies, the PEDro scale was assessed using a 11-item scale that assesses randomized controlled trials from 0 to 1 in each item, where a score of six is the cut-off values for high-quality studies (Bujalance-Moreno, Latorre-Román & García-Pinillos, 2019; García-Pinillos, Soto-Hermoso & Latorre-Román, 2017). Previous research has reported a good test–retest (r = 0.58–0.88) and inter-rater reliability (r = 0.68–0.75) for both qualitative indexes (Downs & Black, 1998; Maher et al., 2003). For this systematic review, the quality assessment was independently performed by two authors (J.E.T, P.F.) with subsequent inter-observer reliability analysis (Kappa index: 0.91; 95% IC [0.90–0.92]).

Study coding and data extraction

Data extraction of the reviewed articles was organized into the following topics: (1) sampling characteristics by the study design, population, competitive level, sample (N), sex, age, expertise level and quality score (Table 2); (2) summary of performance dimension, measures, measurement, thresholds and/or metric formula in the reviewed articles; (3) references and ‘further reading’ reports the original studies where the methodology of the included articles were based; (4) methodological approaches of the reviewed studies by reporting the study purpose, experimental approach, methodological procedures, data collection, statistical and mathematical analysis. Data were collected as previously described in ‘The Cochrane Data Extraction Template for Included Studies’ using a Microsoft Excel sheet (Microsoft Corporation, Readmond, WA, USA (Synnot et al., 2020).

Table 2 Summary of the sampling characteristics in the studies included for systematic review and its quality score.

Reference (year)	Study design	Population, competitive level	Sample (N)	Sex	Age (y)	Expertise level (y)	
Baptista et al. (2020)	RCT	Adult, Semiprofessional	23	Male	24.9 ± 6.5	12.6 ± 5.5	
Canton et al. (2021)	RCT	Youth, High-Level	24	Male	U12: 11.3 ± 0.8	U12: 3.13 ± 1.5	
Coutinho et al. (2017)	RCT	Youth, Amateur	12	Male	15.9 ± 0.8	8.9 ± 2.4	
Coutinho et al. (2019b)	RCT	Youth, Amateur	12	Male	15.9 ± 0.8	8.9 ± 2.4	
Coutinho et al. (2019a)	RCT	Youth, ND	40	Male	U13 (n = 20): 11.3 ± 0.8
U15 (n = 20): 13.3 ± 0.6	U13 (n = 20): 4.9 ± 2.7
U15 (n = 20): 7.0 ± 1.6	
Coutinho et al. (2020)	RCT	Youth, ND	10	Male	13.7 ± 0.5	6.1 ± 0.9	
Coutinho et al. (2022b)	RCT	Youth, ND	114	Male	U9: 7.9 ± 0.9
U11: 9.5 ± 0.9
U13: 11.6 ± 0.8
U15: 13.9 ± 0.6
U17: 16.2 ± 0.7
U19: 17.9 ± 0.4	U9: 2.7 ± 1.1
U11: 3.9 ± 1.2
U13: 4.9 ± 2.0
U15: 6.8 ± 2.5
U17: 7.9 ± 2.8
U19: 9.5 ± 2.1	
Coutinho et al. (2022c)	RCT	Youth, Regional Level	20	Male	16.1 ± 0.9	7.5 ± 3.4	
Coutinho et al. (2022a)	RCT	Youth, ND	21	Male	16.2 ± 0.6	8.3 ± 2.8	
Ferraz et al. (2020)	RCT	Adult, Professional	20	Male	22.3 ± 2.1	10.3 ± 3.4	
Figueira et al. (2018)	RCT	Youth, Elite	22	Male	U15 (n = 22): 13.6 ± 0.4
U17 (n = 22): 15.3 ± 0.4	U15 (n = 22): 5.1 ± 1.3
U17 (n = 22): 7.2 ± 1.4	
Folgado et al. (2015)	Observational cohort	Adult, Professional	23	Male	25.5 ± 3.6	9.0 ± 3.7	
Folgado, Gonçalves & Sampaio (2018)	Observational cohort	Adult, Professional	30	Male	23.7 ± 4.2	4.8 ± 4.2	
Folgado et al. (2019)	RCT	Youth, National Level	20	Male	U15: 14.1 ± 0.5	U15: 6.4 ± 1.8	
Gonçalves et al. (2014)	Observational cohort	Youth, Elite	22	Male	18.1 ± 0.7	9.4 ± 1.3	
Gonçalves et al. (2017)	RCT	Adult, Professional	19	Male	25.1 ± 4.1	18.8 ± 5.3	
Gonçalves et al. (2018a)	Observational cohort	Adult, Professional	28	Male	24.7 ± 4.7	6.5 ± 4.7	
Jara et al. (2019)	RCT	Adult, Elite	3	Male	24.7 ± 7.2	11.0 ± 4.7	
Machado et al. (2022)	RCT	Youth, Recreational	10	Male	16.89 ± 0.11	ND	
Nieto et al. (2022)	RCT	Youth, Elite	22	Male	14.6 ± 0.3	5.5 ± 0.5	
Olthof, Frencken & Lemmink (2018)	Observational cohort	Youth, Professional	148	Male	U13 (n = 36): 12.5 ± 0.5
U15 (n = 43): 14.4 ± 0.5
U17 (n = 28): 16.6 ± 3.2
U19 (n = 43): 17.9 ± 1.0	ND	
Praça et al. (2016)	Observational cohort	Youth, National Level	18	Male	16.4 ± 0.7	4.2 ± 0.0	
Praça et al. (2021)	Observational cohort	Youth, National Level	50	Male	U17 (n = 25): 16.79 ± 0.61
U20 (n = 25): 19.08 ± 0.61	ND	
Ric et al. (2016a)	RCT	Adult, Professional	8	Male	26 ± 4.96	19.6 ± 4.9	
Ric et al. (2017)	RCT	Adult, Professional	21	Male	25.1 ± 4.1	18.8 ± 5.3	
Sampaio et al. (2014)	Observational cohort	Adult, volunteer	24	Male	20.8 ± 1.0	5.2 ± 1.3	
All studies	–	–	764	–	16.81 ± 1.63	4.2 ± 3.83	
Note:

Abbreviations: ND, Not described; U, Under; QS, Quality Score; RCT, randomized controlled trial; y, years.

Results

Search results and study selection

A total of 982 titles were collected on three database (WoS = 495; Pub-Med = 232 and SportDiscus = 255). After applying the selection criteria, 153 full-text articles were screened for eligibility, having 26 articles been retained for final review. Figure 1 shows PRISMA flow diagram depicting the screening procedures and search results.

Figure 1 PRISMA flow diagram.

Participant characteristics

The reviewed articles were published between 2000 and 2022. Sample sizes ranged between 8–148 participants with an observational, prospective and cross-sectional design (n = 8) and randomized controlled trial (n = 18). Twenty-three articles focused on adult football players and seven on youth counterparts. All articles studied male football players, particularly in elite (n = 2), professional (n = 8), high-level (n = 1), national level (n = 2), amateur (n = 2) and volunteer (n = 1) performers. A total of 538 football players were analysed in this systematic review. Age and expertise level were 16.81 ± 1.63 and 4.2 ± 3.83 years, respectively. Table 2 provides the demographic characteristics of the participants in the retained studies.

Quality assessment

In the evaluation of methodological quality, the qualitative scores for cross-sectional studies ranged from 8 (lowest quality) to 11 (highest quality) out of a maximum of 14 possible points in the Downs and Black scale (Table 3). For intervention studies, the PEDro score ranged between six (lowest quality) and nine (highest quality) out of 11 points (Table 4).

Table 3 Modified downs and black scale for reviewed intervention studies.

Reference (year)	Item
1	Item
2	Item
3	Item
6	Item
7	Item
10	Item
12	Item
15	Item
16	Item
18	Item
20	Item
22	Item
23	Item
25	Total score
(out of 14)	
Folgado et al. (2015)	1	1	1	1	0	1	1	0	1	0	1	1	0	0	9	
Folgado, Gonçalves & Sampaio (2018)	1	1	1	0	1	1	1	0	0	1	1	1	1	0	10	
Gonçalves et al. (2014)	1	1	1	1	0	1	1	1	1	0	0	0	0	1	8	
Gonçalves et al. (2018a)	1	1	1	1	1	0	1	0	1	1	1	0	1	1	11	
Olthof, Frencken & Lemmink (2018)	1	1	1	1	0	1	1	1	1	0	0	1	0	1	10	
Praça et al. (2016)	1	1	1	1	0	1	1	0	1	1	0	0	1	0	9	
Praça et al. (2021)	1	1	1	1	1	1	1	1	1	0	0	1	0	1	11	
Sampaio et al. (2014)	1	1	0	1	1	0	1	0	1	1	0	0	1	0	8	
Note:

0 = no; 1 = yes; U = unable to determine. Item 1: clear aim/hypothesis; Item 2: outcome measures clearly described; Item 3: patient characteristics clearly described; Item 6: main findings clearly described; Item 7: measures of random variability provided; Item 10: actual probability values reported; Item 12: participants prepared to participate representative of entire population; Item 15: blinding of outcome measures; Item 16: analysis completed was planned; Item 18: appropriate statistics; Item 20: valid and reliable outcome measures; Item 22: participants recruited over same period; Item 23: randomised; Item 25: adjustment made for confounding variables.

Table 4 Physiotherapy evidence database scale (PEDro) for reviewed intervention groups.

Reference (year)	Item
1	Item
2	Item
3	Item
4	Item
5	Item
6	Item
7	Item
8	Item
9	Item
10	Item
11	Total score
(out of 11)	
Baptista et al. (2020)	1	1	1	1	1	1	1	0	1	1	0	9	
Canton et al. (2021)	1	1	1	1	1	0	1	1	1	0	0	8	
Coutinho et al. (2017)	1	1	1	1	1	1	1	1	1	0	0	9	
Coutinho et al. (2019b)	1	1	1	1	1	1	1	0	1	1	0	9	
Coutinho et al. (2019a)	0	1	1	1	1	0	0	1	1	0	0	6	
Coutinho et al. (2020)	1	1	1	0	1	0	1	1	1	0	1	8	
Coutinho et al. (2022b)	1	1	1	1	1	1	1	0	1	1	0	9	
Coutinho et al. (2022c)	0	1	1	1	1	1	1	0	1	0	1	8	
Coutinho et al. (2022a)	1	1	1	1	1	1	1	0	1	0	1	9	
Ferraz et al. (2020)	1	1	0	1	1	1	1	1	1	0	0	8	
Figueira et al. (2018)	1	1	1	1	1	1	1	1	1	0	0	9	
Folgado et al. (2019)	1	1	1	1	1	1	1	1	1	0	0	9	
Gonçalves et al. (2017)	1	1	1	1	1	0	1	0	1	0	0	7	
Jara et al. (2019)	1	1	1	1	0	1	1	1	1	0	0	8	
Machado et al. (2022)	1	1	1	1	1	1	1	0	1	0	0	8	
Nieto et al. (2022)	1	1	1	1	0	1	1	0	0	1	0	7	
Ric et al. (2016a)	1	1	1	1	1	1	1	0	1	1	0	9	
Ric et al. (2017)	1	1	1	1	1	0	1	0	1	0	0	7	
Note:

0 = Item was not satisfied; 1 = item was satisfied. Item 1: eligibility criteria were specified; Item 2: subjects were randomly allocated to groups; Item 3: allocation was concealed; Item 4: the groups were similar at baseline regarding the most important prognostic indicators; Item 5: there was blinding of all subjects; Item 6: there was blinding of all therapists who administered the therapy; Item 7: there was blinding of all assessors who measured at least one key outcome; Item 8: measurements of at least one key outcome were obtained from more than 85% of the subjects initially allocated to groups; Item 9: all subjects for whom outcome measuments were available received the treatment or control condition as allocated , or where this was not the case, data for at least one key outcome were analysed by “intention to treat”; Item 10: the results of between groups statistical comparisons are reported for at least one key outcome; Item 11: the study prevides both point measurements and measurements of variability for at least one key outcome.

Main findings

Table 5 presents the data extraction of the retained studies. Concerning the physical data, external training load measures selected were based on movement speed, specifically: (i) total distance covered (n = 11), (ii) distance covered at different speed zones (n = 13), (iii) game pace or average speed (n = 3), (iv) accelerations and decelerations (n = 3), (v) locomotive-based ratios (e.g., ratio between the distance covered at different intensities and distance) (n = 1). Otherwise, positional and tactical variables reported in the included studies were based on the following independent variables: (i) possession ball (n = 1), (ii) spatial exploration indexes (n = 6), (iii) LPW ratio (n = 1), (iv) stretch indexes (n = 2), (v) multiscale entropy (n = 1), (vi) synchronization indexes (i.e., longitudinal and lateral directions) (n = 10), (v) intra-team and opponent’s dyads (n = 3), (vi) dispersion and contraction indexes (i.e., length, width and speed) (n = 4), (vii) playing space and effectiveness (i.e., effective playing space, longitudinal distance between GK and the closest defender (n = 3), (viii) player’s variability, regularity and coordination (n = 10) (i.e., entropy, dynamic overlap, near-in-phase and near-anti-phase coordination, regularity zones occupied), (ix) team centroid (n = 2).

Table 5 Summary of performance dimensions, measures, measurements and their thresholds/metric formulas in the included articles.

Dimension	Measure		Measurement	Description, thresholds and/or metric formula	Reference	Further reading	
Physical data	External load	Movement speed	TD (m)	Higher ratio ( >16 km·h−1), moderate ratio (10.0–15.9 km·h−1), lower ratio: 7.0–9.9 km·h−1) with distance covered at very low intensities (0.0–6.9 km·h−1)	(Canton et al., 2021; Coutinho et al. 2019b, 2019a)	(Abade et al., 2014; Bradley & Ade, 2018; Hodgson, Akenhead & Thomas, 2014)	
Walking (0.0–3.5 km·h−1), jogging (3.6–14.3 km·h−1), running (14.4–19.7 km·h−1), and sprinting (>19.8 km·h−1).	(Coutts & Duffield, 2010; Ferraz et al., 2020; Figueira et al., 2018; Folgado et al., 2015; Folgado, Gonçalves & Sampaio, 2018; Gonçalves et al., 2017, 2018a; Ric et al., 2016a, 2017)	(Duarte et al., 2013b; Folgado et al., 2014a; Giménez et al., 2018; Gonçalves et al., 2017)	
Zone 1 (0–6.9 km·h−1); zone 2 (7–9.9 km·h−1); zone 3 (10–12.9 km·h−1); zone 4 (13–15.9 km·h−1); zone 5 (16–17.9 km·h−1) and zone 6 (≥18 km·h−1).	(Sampaio et al., 2014)	(Hill-Haas et al., 2008)	
High speed (km·h−1)	Distance covered in the high ratio/distance covered in walking multiplied by 100.	(Coutinho et al., 2019b, 2019a)	(Abade et al., 2014; Gonçalves et al., 2018b)	
Distance covered at high intensity (≥19.8 km·h−1) and number of sprints (frequency of displacements ≥25.2 km·h-1)	(Olthof, Frencken & Lemmink, 2018)	(Abt & Lovell, 2009; Goto, Morris & Nevill, 2015)	
Distance covered at three speed zones (14.40–19.79 km·h−1, 19.80–22.99 km·h−1, higher than 23.00 km·h−1) and number of sprints (frequency of displacements ≥23.00 km·h−1)	(Praça et al., 2021)	(Mallo et al., 2015; Praça et al., 2020)	
Game pace or average speed (km·h−1 or CV)	Players’ average speed displacement, expressed as meters or CV.	(Canton et al., 2021; Coutinho et al., 2019b, 2019a)	(Ferraz et al., 2017, 2018; Gonçalves et al., 2019)	
ACC/DEC (m·s−2)	ACC: 0.5–3.0 m·s−2; DEC: > −3.0 0 m·s−2	(Coutinho et al., 2017)	(Dalen et al., 2016; Russell et al., 2016)	
Body load	(ay1−ay−1)2+ax1−ax−1)2+(az1−az−1))2100	(Gonçalves et al., 2017)	(Buchheit et al., 2014)	
Positional data	Spatial and temporal features	Spatial movement variability/regularity	CV	Magnitude of the variability in the distance between players’, expressed by the coefficient of variation CV (%)	(Coutinho et al., 2019b; Coutinho et al., 2020; Figueira et al., 2018; Gonçalves et al., 2017)	(Frencken et al., 2012; Harbourne & Stergiou, 2009; Seifert, Button & Davids, 2013; Travassos et al., 2014)	
			ApEn	Ranged 0 to 2, in which lower values correspond to more repeatable patterns). The imputed values used to compute were 2 to vector length (m) and 0.2*std to the tolerance (r).	(Baptista et al., 2020; Coutinho et al., 2020; Coutinho et al., 2019b; Ferraz et al., 2020; Figueira et al., 2018; Gonçalves et al., 2017; Sampaio et al., 2014)	(Duarte et al., 2012b; Gonçalves et al., 2016; Gréhaigne, Bouthier & David, 1997; Pincus, 1991; Preatoni et al., 2010; Richman & Moorman, 2000; Seifert, Button & Davids, 2013; Silva et al., 2016b; Stergiou et al., 2004; Yentes et al., 2013)	
			Boltzmann–Gibbs–Shannon entropy	Probabilities of configurations were calculated as limit (large N) relative frequencies for stationary distributions: pi = ni/N where ni and N is the frequency and number of the configuration respectively.	(Ric et al., 2016a)	(Balescu, 1975)	
			MSE
SamEn	SampEn and MSE curves to a range of different timescales, calculating the area under and complexity index.	(Canton et al., 2021)	(Busa & van Emmerik, 2016; Costa, Goldberger & Peng, 2005)	
		Complex index	SEI	Width and length displacements from each positioning time series to the mean position.	(Canton et al., 2021; Figueira et al., 2018; Gonçalves et al., 2017; Praça et al., 2021)	(Arede et al., 2021; Clemente et al., 2018a, 2018b; Travassos et al., 2014)	
			Stretch index (meters or ApEn)	Men of the distances between each player and the geometric centre of the team.	(Coutinho et al., 2019b; Olthof, Frencken & Lemmink, 2018; Praça et al., 2021)	(Bourbousson, Sève & McGarry, 2010; Clemente et al., 2013; Clemente et al., 2018b; Duarte et al., 2013b; Lames, Erdmann & Walter, 2010; Travassos et al., 2014)	
			Dynamic overlap <qd(t)>	Average cosine auto-similarity of the overlap between configurations with increasing time lag:
⟨qd(t)⟩=(1−qstat)tα+qstat	(Ric et al., 2016a)	(Hristovski et al., 2013; Saxton, 1996)	
			Trapping strength	Probability of remaining inside the same attractor that is a conditional probability of a configuration being subsequently repeated (i.e., trapping strength and behavioural flexibility).	
			Voronoi algoritms	Voronoi algorithms allow to compute a diagram represented by spatial cells for individual positional area (m2).	(Baptista et al., 2020)	(Fonseca et al., 2012)	
		Coordination/
synchronization
using intra-team dyads	Relative phase
(Hilbert transform)	Longitudinal and lateral directions using near-in-phase synchronization of each dyad that was quantified by the percentage of time spent between −30° to 30° bin.	(Coutinho et al., 2020; Coutinho et al., 2019a; Fernandes et al., 2010; Figueira et al., 2018; Folgado et al., 2015, 2019; Gonçalves et al., 2018b, 2017)	(Duarte et al., 2012b, 2012c, 2013a; Folgado et al., 2014a, 2014b; Gonçalves et al., 2019; McGarry et al., 2002; Palut & Zanone, 2005; Sampaio & Maçãs, 2012; Silva et al., 2016a; Travassos et al., 2011a; 2013)	
Speed synchronisation	0.0–3.5 km·h−1 (low intensity); 3.6–14.3 km·h−1 (moderate intensity); 14.4–19.7 km·h−1 (high intensity); and >19.8 km·h−1 (very high intensity).	(Folgado, Gonçalves & Sampaio, 2018; Gonçalves et al., 2018a)	(Folgado et al., 2014b)	
Distance player–teammate	Interpersonal distance between each pair of players, both with teammates and opponents:
D(ax(t),y(t),bx(t),y(t))=(ax(t),y(t))2+(ax(t),y(t))2

where D is the distance, a is the player, x and y are the coordinates, and t is the time, and b is the teammate or opponent.	(Ferraz et al., 2020; Olthof, Frencken & Lemmink, 2018; Ric et al., 2017; Sampaio et al., 2014)	(Gonçalves et al., 2014; Low et al., 2019; Silva et al., 2016c)	
Coordination/synchronization using inter-team dyads	Distance player–opponent	
Playing space	Distance from the target	Distance from the target according to ten categories: >37.45 m; 32.1 ± 37.45 m; 36.75 ± 32.1 m; 21.4 ± 26.75 m; 16.05 ± 21.4 m; 10.7 ± 16.05 m; 5.35 ± 10.7 m; 0 ± 5.35 m.	(Ric et al., 2017)	(Duarte et al., 2012b)	
Total surface area or team effective playing space (m2)	Smallest convex hull, that is the smallest polygonal area that it is
delimited by the peripheral players	(Coutinho et al., 2017; Folgado et al., 2019; Olthof, Frencken & Lemmink, 2018)	(Duarte et al., 2013b; Folgado et al., 2014b; Mendes, Malacarne & Anteneodo, 2007; Ric et al., 2016b; Russell et al., 2016; Sampaio & Maçãs, 2012)	
Ellipses: SEA and PEA areas	Spatial analysis for a set of points in a two-dimensional space, which boundaries will enclose about the 100 (1 – α):

x¯=1n∑i=0n⁡xi,y¯=1n∑i=1n⁡yi
	(Jara et al., 2019)	(Batshcelet, 1981; Lefever, 1926; Yuill, 1971)	
Team’s width and length	Longitudinal position of team geometrical center (x axis) and lateral position of team geometrical center (y axis), expressed as meters, CV and length-per-width (LPW) ratio per team.	(Baptista et al., 2020; Canton et al., 2021; Coutinho et al., 2017; Gonçalves et al., 2017; Praça et al., 2021)	(Duarte et al., 2013b; Folgado et al., 2014b; Frencken et al., 2011; Mendes, Malacarne & Anteneodo, 2007; Ric et al., 2016a)	
Team’s speed contraction dispersion	
Team centroid	
Other dimensions
(non-positional data)	Technical variables	Individual actions	Passes, dribbles and shots	Successful passes (%), successful dribbles (%), shots on target (%), goals (%)	(Coutinho et al., 2020)	(Liu et al., 2016; O’Donoghue, 2009; Santos et al., 2018)	
Distance covered at different intensities when dribbling; number of completed passes; completed passes distance; shots distance to the goal; distance between attacker and defender when shooting.	(Folgado et al., 2019; Olthof, Frencken & Lemmink, 2018)	(Rampinini et al., 2009)	
Successful pass reception and turnovers, goals scored and relative frequencies of players’ passing interactions.	(Ric et al., 2017)	(Costa, Goldberger & Peng, 2005)	
Transition probabilities were calculated dividing the number of each player’s passes to his teammates, turnovers and goals by the total number of player interactions.	
	Tactical variables	Ball possession	Offensive/defensive
Phases	Duration of possession, team width, team length and their ratio
(LPWR), as well as their coefficient of variation	(Canton et al., 2021; Olthof, Frencken & Lemmink, 2018; Ric et al., 2016a)	(Collet, 2013; Costa et al., 2011; Gabin et al., 2012)	
Network	Dyad nodes	Relative phase analysis was also divided according to each dyad average speed in three levels: for the whole team; for dyads with similar synchronisation tendencies; and for each dyad.	(Folgado et al., 2015; Ric et al., 2017)	(McGarry et al., 2002; Travassos et al., 2011a, 2013)	
Tactical actions	Patterns/categories	Tactical actions classified as: penetration, offensive coverage, depth mobility, width and length, offensive unity, delay, defensive coverage, balance, concentration, defensive unity.	(Ric et al., 2017)	(Costa, Goldberger & Peng, 2005)	
Psycho-physiological variables	Perceived exertion	RPE (a.u.)	CR10-scale (0 to 10 arbitrary units).	(Coutinho et al., 2017)	(Lee, Hicks & Nino-Murcia, 1991)	
	Exertion index	Wisbey’s formula: players’ instantaneous speed (over 10 s and speed over 60 s).	(Folgado et al., 2015)	(Wisbey et al., 2010)	
	Heart rate	HRmax (bpm)	Percentage of HRmax into intensity zones: Zone 1 (<75% HRmax), Zone 2 (75–84.9% HRmax), Zone 3 (85–89.9% HRmax), and Zone 4 (≥90% HRmax).	(Folgado, Gonçalves & Sampaio, 2018; Sampaio et al., 2014)	(Abt & Lovell, 2009; Gore, 2000)	
	Average HR (bpm)	Average beats per minute (BPM)	(Gonçalves et al., 2017)	ND	
	TRIMPMOD	Total TRIMPMOD: zone 1 (65–71% HRmax) * 1.25; zone 2 (72–78% HRmax) * 1.71; zone 3 (79–85% HRmax) * 2.54; zone 4 (86–92% HRmax) * 3.61; and zone 5 (93–100% HRmax) * 5.16	(Folgado, Gonçalves & Sampaio, 2018)	(Campos-Vazquez et al., 2015; Los Arcos et al., 2014; Stagno, Thatcher & van Someren, 2007)	
Note:

Abbreviations: <qd(t)> – Dynamic overlap; ACC/DEC – Accelerations and decelerations; ApEn – Approximate entropy; BPM – Average beats per minute; CR-10 – Borg CR10 scale; CV – Coefficient of variation; D – Distance; HR – Heart Rate; HRmax – maximum Heart Rate; LPWR – Team width, team length and their ratio or LPW – length-per-width ratio per team; MSE – Multiscale Entropy; PEA – Prediction Ellipse; RPE – Ratings of Perceived Exertion; SamEn – Sample Entropy; SEA – Standard Ellipse; SEI – Spatial exploration index; TD – Total distance; TRIMP – Training Impulse; TRIMPMOD – modified Training Impulse.

Table 6 lists the purpose, game format, experimental approach, methodological procedures, data collection, statistical and mathematical analysis of the studies included in this review. The data organization respected the main purposes of this systematic review, specifically the integration of physical and tactical variables in football using positional data (n = 26). Other dimensions were also reported, such as psychological and technical factors. However, the results of these approaches were not the focus of the analyses (n = 5). Non-positional variables were computed by the reviewed studies for other performance dimension such as technical (n = 4), tactical (n = 5) and psychophysiological variables (n = 4). Psychophysiological measures were reported by exertion-based indexes (i.e., exertion index per minute, ratings of perceived exertion) (n = 1) and heart rate-based methods (i.e., %HRmax, TRIMPMOD) (n = 3).

Table 6 Methodological approaches of included articles.

Referente (year)	Study purpose	Experimental approach	Methodological procedures	Data collection (Device specification)	Statistical and mathematical analysis	
Match-play	Training set	Game format	Physical/
physiological	Positional/
tactical	Other dimensions	
Baptista et al. (2020)	Identified the effects of playing formations on tactical behaviour and external workload during SSG.	✗	SSG	GK + 7 vs 7 + GK	✓	✓	✗	MATLAB® routines (MathWorks, Inc., Natick, MA, USA)	Non-differential 5 Hz GPS (ND)	Cohen’s d
SWC
MBI
ApEn
Voronoi algorithme	
Canton et al. (2021)	Identified how positioning the goals in diagonal configurations on the pitch modifies the external training load and the tactical behaviour during SSG.	✗	SSG	GK + 5 vs 5 + GK	✓	✓	✗	MATLAB® routines (MathWorks, Inc., Natick, MA, USA)
Lince software® (Gabin et al., 2012),	10 Hz GPS units (WIMU PRO, RealTrack Systems, Almeria, Spain)	Cohen’s d
SWC
MBI
NHST	
Coutinho et al. (2017)	Examined the effects of mental fatigue and additional corridor and pitch sector lines on players’ physical and tactical performances during SSG.	✗	SSG	GK + 6 vs 6 + GK	✓	✓	✓	MATLAB® routines (MathWorks, Inc., Natick, MA, USA)	CR10-scale (RPE)
Portable optical timing system (Optojump, Microgate, Bolzano, Italy)
15 Hz GPS (SPIPRO, GPSports, Canberra, ACT, Australia)	Cohen’s d
SWC
NHST	
Coutinho et al. (2019b)	Identified the effects of adding spatial references during SSG on players’ tactical and physical performance.	✗	SSG	GK + 6 vs 6 +GK	✓	✓	✓	MATLAB® routines (MathWorks, Inc., Natick, MA, USA)	5 Hz GPS (SPI-PRO, GPSports, Canberra, ACT, Australia)	CV
ApEn
MBI
NHST	
Coutinho et al. (2019a)	Identified the effects of different pitch configurations on players’ positional and physical performance.	✗	SSG	GK + 5 vs 5 + GK	✓	✓	✓	MATLAB® routines (MathWorks, Inc., Natick, MA, USA)	5 Hz GPS (SPI-PRO, GPSports, Canberra, ACT, Australia)	CV
ApEn
MBI
NHST	
Coutinho et al. (2020)	Compared players’ performances when manipulating the external markings of the pitch during SSG.	✗	SSG	GK + 5 vs 5 + GK	✓	✓	✓	MATLAB® routines (MathWorks, Inc., Natick, MA, USA)
LongoMatch software (Longomatch, version 1.3.7., Fluendo)	5 Hz GPS (SPI-PRO, GPSports, Canberra, ACT, Australia)
Digital video camera (Sony NV-GS230)	CV
ApEn
MBI
NHST	
Coutinho et al. (2022b)	Explored how the number of allowed ball touches per player possession affected the performance of different age groups during SSG.	✗	SSG	GK + 6 vs 6 +GK	✓	✓	✓	MATLAB® routines (MathWorks, Inc., Natick, MA, USA)
LongoMatch software (Longomatch, version 1.3.7., Fluendo	5 Hz GPS (SPI-PRO, GPSports, Canberra, ACT, Australia)	Cohen’s d
SWC
NHST
Hilbert transform	
Coutinho et al. (2022c)	Aimed to identify the effects of playing with additional individual, collective or individual-collective variability on players’ performance during SSG.	✗	SSG	GK + 6 vs 6 +GK	✓	✓	✓	MATLAB® routines (MathWorks, Inc., Natick, MA, USA)
LongoMatch software (Longomatch, version 1.3.7., Fluendo	5 Hz GPS (SPI-PRO, GPSports, Canberra, ACT, Australia)	Cohen’s d
NHST
Hilbert transform	
Coutinho et al. (2022a)	Explored how manipulating the colour of training vests affects footballers’ individual and collective performance during SSG.	✗	SSG	GK + 6 vs 6 +GK	✓	✓	✓	MATLAB® routines (MathWorks, Inc., Natick, MA, USA)
LongoMatch software (Longomatch, version 1.3.7., Fluendo	5 Hz GPS (SPI-PRO, GPSports, Canberra, ACT, Australia)	Cohen’s d
NHST
Hilbert transform	
Ferraz et al. (2020)	Identified how the manipulation of knowledge regarding a training task duration constrains the pacing and tactical behaviour in SSG.	✗	SSG	GK + 5 vs 5 + GK	✓	✓	✓	ND	5 Hz GPS (SPI-Pro X II, GPS ports, Canberra, ACT, and Australia)	ApEn
MBI	
Figueira et al. (2018)	Compared footballers’ performances when playing with teammates and opponents from the same or different age groups	✓	–	GK + 11 vs 11 + GK	✓	✓	✗	MATLAB® routines (MathWorks, Inc., Natick, MA, USA)	5 Hz GPS (SPI-Pro X II, GPS ports, Canberra, ACT, and Australia)	CV
ApEn
MBI	
Folgado et al. (2015)	Examined the physical and tactical performances under congested and non-congested fixture periods	✓	–	GK + 11 vs 11 + GK	✓	✓	✗	MATLAB® routines (MathWorks, Inc., Natick, MA, USA)	Semiautomatic tracking system (Prozone®, ProZone Holdings Ltd, Leeds, UK).	CHAID
Cohen’s d
NHST
Hilbert transform	
Folgado, Gonçalves & Sampaio (2018)	Identified changes in tactical, physical and physiological performances in LSG during the preseason.	✗	LSG	GK + 8 vs. 8 + GK	✓	✓	✗	MATLAB® routines (MathWorks, Inc., Natick, MA, USA)	5 Hz GPS (SPI-Pro X II, GPS ports, Canberra, ACT, and Australia)	NHST
Cohen’s d	
Folgado et al. (2019)	Compared players’ performance during two SSG with different pitch orientation (i.e., 40 × 30 m vs. 30 × 40 m).	✗	SSG	GK + 4 vs 4 + GK	✓	✓	✗	MATLAB® routines (MathWorks, Inc., Natick, MA, USA)	10 Hz GPS (MinimaxX S5; Catapult Innovations, Docklands, Australia).
Digital Vídeo Camera (Canon PowerShot SX720 HS; Canon Inc, Tokyo, Japan),	ICC
SEM
Cohen’s d
SWC
MBI	
Gonçalves et al. (2014)	Identified differences in time–motion, modified training impulse, body load and movement behaviour between defenders, midfielders and forwards, during an 11-a-side simulated football game.	✓	–	GK + 11 vs 11 + GK	✓	✓	✗	MATLAB® routines (MathWorks, Inc., Natick, MA, USA)	5 Hz GPS (SPI-Pro X II, GPS ports, Canberra, ACT, and Australia)
1 Hz short-range radio telemetry (Polar Team Sports System, Polar Electro Oy, Finland)	Hilbert transform
ApEn
NHST	
Gonçalves et al. (2017)	Identified how pitch area restrictions affect the players’ tactical behavior, physical, and physiological performances during LSG.	✗	LSG	GK + 9 vs. 9 + GK GK+ 10 vs. 10 + GK	✓	✓	✗	MATLAB® routines (MathWorks, Inc., Natick, MA, USA)	5 Hz GPS (SPI-Pro X II, GPS ports, Canberra, ACT, and Australia)
1 Hz short-range radio telemetry (Polar Team Sports System, Polar Electro Oy, Finland)	Hilbert transform
CV
ApEn
Cohen’s d
SWC
MBI	
Gonçalves et al. (2018a)	Examined the changes in the players’ speed synchronization and physical performance between the first and the second half (15-min time). Explored the match-to-match variation of players’ speed synchronization performance.	✓	–	GK + 11 vs 11 + GK	✓	✓	✗	MATLAB® routines (MathWorks, Inc., Natick, MA, USA)	Match Analysis Camera System®.	Hilbert transform
CV
Cohen’s d
SWC
MBI	
Jara et al. (2019)	Analyzed how the modification of the pitch size in SSGs affects the GK’s physical demands.	✗	SSG
MSG
LSG	ND	✓	✓	✗	MATLAB® routines (MathWorks, Inc., Natick, MA, USA)	18.18 Hz GPS (GPEXE GK, Exelio SRL, Udine, Italy)	Cohen’s d
MBI
NHST	
Machado et al. (2022)	Investigated how different strategies of task constraint manipulation impact physical and tactical demands in small-sided and conditioned games (SSCG)	✗	SSG	GK + 4 vs 4 + GK	✓	✓	✗	MATLAB® routines (MathWorks, Inc., Natick, MA, USA) and SPROTM (RealTrack System, Almería, Spain)	10 Hz GPS and inertial devices (WIMU ProTM and GPS, RealTrack System, Almería, Spain)	Effect size (ND)
NHST	
Nieto et al. (2022)	Described the effects on player’s collective behaviour and physical response in three different pitch lengths (100, 75 and 50 m) keeping the width constant (60 m)	✗	LSG	GK + 11 vs 11 + GK	✓	✓	✗	Microsoft Excel Visual Basic for Applications (VBA) (Microsoft,
Redmond, WA, USA)	10 Hz GPS (MinimaxX S5, Catapult Innovations)	Cohen’s d
MBI
NHST
SampEn	
Olthof, Frencken & Lemmink (2018)	Investigated SSGs with a traditional small pitch and a match-derived relative pitch area in youth elite soccer players.	✓	SSG	GK + 4 vs 4 + GK
GK + 11 vs 11 + GK	✓	✓	✗	MATLAB® routines (MathWorks, Inc., Natick, MA, USA)	Two HD video dome cameras (Bosch GmbH., Stuttgart, Germany) and one or two high resolution digital cameras (Canon HF100, Canon Inc., Tokyo, Japan; JVC Everio, JVC Kenwood Corporation, Kanagawa, Japan).
LPM system (Inmotio Object Tracking BV., Amsterdam, The Netherlands)	NHST
Effect sized eta-squared (ηp2)	
Praça et al. (2016)	Compared the collective tactical behavior between numerically balanced and unbalanced SSG.	✗	SSG	3 vs 3
3 vs 3 + 2
4 vs 3	✓	✓	✗	MATLAB® routines (MathWorks, Inc., Natick, MA, USA)	15 Hz GPS (model SPI-Pro X2; GPSports, Canberra, Australia)	NHST
Effect sized eta-squared (ηp2)	
Praça et al. (2021)	Analysed the effects of changing the match venue on match-related player’s physiological, physical, and tactical responses with an age-dependent.	✓	–	GK + 11 vs 11 + GK	✓	✓	✗	MATLAB® routines (MathWorks, Inc., Natick, MA, USA)	10 Hz GPS device, with an embedded 200 Hz accelerometer and 1 Hz heart rate monitor (Polar®, Team Pro, Kempele, Finland).	NHST
Effect sized eta-squared (ηp2)	
(Ric et al., 2016a)	Identified the dynamics of tactical behaviour emerging on different timescales in SSG. Quantified short- and long-term exploratory behaviour according to the number of opponents.	✗	SSG	GK + 4 vs 3 + GK
GK + 4 vs 5 + GK
GK + 4 vs 7 + GK	✓	✓	✗	MATLAB® routines (MathWorks, Inc., Natick, MA, USA)
Lince software® (Gabin et al., 2012)	Digital video camera for video recording and analysed an ad hoc instrument being used to notate tactical actions (Costa et al., 2011)
15 Hz GPS (SPI-ProX, GPS ports, Canberra, ACT, and Australia)	Dynamic overlap <qd(t)>
Trapping strength
Boltzmann–Gibbs–Shannon entropy
NHST
Cohen’s d	
(Ric et al., 2017)	Identified how players’ spatial restrictions affected the exploratory tactical behaviour and constrained the perceptual-motor workspace in ball possession and the inter-player passing interactions.	✗	ND	GK + 10 vs 9 + GK	✓	✓	✗	MATLAB® routines (MathWorks, Inc., Natick, MA, USA)	5 Hz GPS (SPI-Pro X II; GPS ports, Canberra, ACT, and Australia)	Dynamic overlap <qd(t)>	
(Sampaio et al., 2014)	Compared and discriminate the time-motion variables, heart rate and players’ tactical behaviour according to game pace, status and team unbalance.	✗	SSG	GK + 5 vs 5 + GK	✓	✓	✗	MATLAB® routines (MathWorks, Inc., Natick, MA, USA)	5 Hz GPS (SPI-Pro X II; GPS ports, Canberra, ACT, and Australia)	NHST
SC	
Note:

Abbreviations: <qd(t)> – Dynamic overlap; ApEn – Approximate entropy; CHAID – Chi-squared automatic interaction detection; Cohen’s d – Standardized (Cohen) differences; CV – Coefficient of variation; GK – Goalkeeper; ICC – Intraclass correlation; LSG – Large-sided games; MBI – Magnitude-based inferences; MSG – Medium-sided games; ND – Not described; NHST – Null hypothesis statistical test; ηp2 – Effect sized eta-squared; SampEn – Sample Entropy; SC – Structural coefficients; SEM – Standard error of measurement; SSG – Small-sided games; SWC – Smallest worthwhile changes; USA – United States; VSA – Visual Basic for Applications.

Quasi-experimental approaches studied training sets (n = 20) and match settings (n = 6). One study analysed both training and play settings. Small sided-games (SSG) were the most common training task formats (n = 17), with only three articles focusing on medium-sided (MSG) (n = 1) and large-sided games (LSG) (n = 4). Regarding the methodological procedures, Matlab® routines (MathWorks, Inc., Natick, MA, USA) were used by all authors for processing raw data (xx, yy) (n = 26). All studies applied Butterworth low pass filter at sampling frequencies ranging 3–5 Hz, using 10–20 windows and, 1,000–3,000 points per data collect. Match analysis software was used to extract technical variables in three studies, including the LongoMatch® software (n = 1), Match Analysis Camera Systems® (n = 1) and Lince software (n = 1). Data collection was based on GPS (n = 15), LPM (n = 2) and optical-based tracking systems (n = 5) at 5–15 Hz. Also, internal training load measures were collected at 1 Hz short-range radio telemetry (n = 2) and CR 10-scale (n = 1). A study used a portable optical timing system to measure neuromuscular performance (i.e., countermovement jump, CMJ).

Null hypothesis statistical test (NHST) and magnitude-based inferences (MBI) were the statistical procedures chosen in seven (n = 7) and five (n = 5) studies, respectively. The statistical and mathematical analyses performed were the approximate entropy (ApEn) (n = 7), Boltzmann–Gibbs–Shannon entropy measure (n = 7), Coefficient of variation (CV) (n = 7), dynamic overlap (<qd(t)>) (n = 7), effect sized Cohen’s d (n = 7), effect sized eta-squared (ηp2) (n = 7), hilbert transform (n = 7), intraclass correlation (ICC) (n = 7), smallest worthwhile changes (SWC) (n = 7), standard error of measurement (SEM) (n = 7), standardized (Cohen) differences (n = 7), structural coefficients (SC) (n = 7), and trapping strength (n = 7).

Discussion

The aim of this study was to systematically review the articles that integrated physical and tactical variables using positional data in football. Physical data used to be analysed by the player’s speed. Otherwise, positional datasets were computed by spatiotemporal features such as spatial variability or regularity of the player’s movements, complex index, coordination/synchronization using intra-team and inter-team dyads, playing space.

Positional datasets allows a more ecological insight on individual physical demands, if the data interpretation considers the contextual factors and collective behaviour through a tactical analysis (Marcelino et al., 2020; Teixeira et al., 2022a; Teixeira et al., 2021c). Several authors have emphasized the need to expand the evidence produced in football on just one performance dimension (i.e., physical/physiological, technical or tactical). It is important to apply methodologies based on integrative approaches that analyse the interplay between technical factors, key tactical/performance outcomes, collective behaviour and match-related contextual drivers (Teixeira et al., 2022b). Therefore, an integrative approach was expanded in eight articles by adding psychophysiological and technical outcomes (Bradley & Ade, 2018; Paul, Bradley & Nassis, 2015). Considering the multifactorial phenomenon of performance in team sports, it is also important to consider the influence of psychological variables on the control of physical capacities, pacing behaviour, decision-making, self-regulation, and effort perception (Branquinho et al., 2020, 2021; Ferraz et al., 2022). Also, bringing together observational methodologies should be considered when positional data is to be made meaningful with skilled and technical aspects (Anguera & Hernández Mendo, 2013; Preciado et al., 2019; Sarmento et al., 2018).

The reviewed quasi-experimental studies researched training sets and match settings. Small sided-games (SSG) was the most common training task in the studies, with only three articles addressing medium-sided (MSG) and large-sided games (LSG) (Machado et al., 2020; Folgado, Gonçalves & Sampaio, 2018; Gonçalves et al., 2017; Jara et al., 2019; Nieto et al., 2022; Praça et al., 2016), respectively. Thus, SSG formats have been further explored in the literature, mainly five- and six-sided game formats. Indeed, these SSG-based formats were previous reported as useful tools to promote significant variations in the training load, and likely in the improvement of the different domains of football training (i.e., physiological, technical, and tactical dimensions) (Branquinho, Ferraz & Marques, 2021; Clemente, Afonso & Sarmento, 2021). Also, SSG and conditioned games (SSCG) are excellent ways to enhance acquisition of motor efficiency and decision-making skills (Davids et al., 2013). A research gap remains unexplored in MSG and LSG formats, as well as, its relationship with formal game formats (i.e., 7-, 8-, 9- and 11-sided formats) (Baptista et al., 2020; Coutinho et al., 2019b, 2020; Ferraz et al., 2020; Figueira et al., 2018; Gonçalves et al., 2017; Sampaio et al., 2014).

Regarding the methodological procedures, MATLAB® routines (MathWorks, Inc., Natick, MA, USA) were employed to process raw data (xx, yy) (n = 26), transforming data points into the Universal Transverse Mercator (UTM) coordinate system (Folgado et al., 2014b; Sampaio & Maçãs, 2012). The most used correction guideline to reduce tracking signal noise was a 3 Hz Butterworth low pass filter by applying non-linear logarithms using 20 windows of 3,000 points per dataset (Coutinho et al., 2019a, 2020; Figueira et al., 2018; Folgado et al., 2015, 2019; Gonçalves et al., 2017, 2018b). Other studies adopted smaller data windows such as 1,500 data sets and a sampling frequency for signal correction (i.e., 5 Hz) (Baptista et al., 2020). Nevertheless, the sampling frequency and datasets is highly dependent on the type of non-linear method to be used, and the use of higher time-series lengths can increase the consistency of the positional data (Baptista et al., 2020; Richman & Moorman, 2000; Yentes et al., 2013). Approximate entropy (ApEn) was the most noted non-linear variable for measuring the spatial movement variability/regularity (Baptista et al., 2020; Coutinho et al., 2019b, 2020; Ferraz et al., 2020; Figueira et al., 2018; Gonçalves et al., 2017; Sampaio et al., 2014). Also, stretch index can be based on the ApEn or distance, being the most reported complex index in the reviewed literature (Coutinho et al., 2019a; Olthof, Frencken & Lemmink, 2018; Praça et al., 2021). Indeed, the entropy has been extensively reported as outstanding informational parameter to describe the variability and predictability of the players’ movements (Teixeira et al., 2022b). Hilbert transform was the most frequent method, by computing the longitudinal and lateral directions through in- and anti-phase (Coutinho et al., 2019a, 2020; Fernandes et al., 2010; Figueira et al., 2018; Folgado et al., 2015, 2019; Gonçalves et al., 2017, 2018b). Total surface area or playing space (m2) can be provided through trigonometry using the smallest convex hull and/or polygonal area delimited by the peripheral players (Coutinho et al., 2017; Folgado et al., 2019; Olthof, Frencken & Lemmink, 2018). Recently, Teixeira et al. (2022b) reported that it remains to be explored correlation matrixes, clustering methods and fractality patterns. This review confirms this assertion and opens up the possibility of exploring these metrics by integrating physical demands with individual and collective behaviour.

Match analysis software was used in three studies for notational analysis to extract technical variables, specifically the LongoMatch® software, Match Analysis Camera Systems® (Gonçalves et al., 2018a) and Lince software® (Canton et al., 2021; Ric et al., 2016b). Further tactical variables were selected in the retained studies using other methodological approaches (i.e., observational and notational analysis) such as metrics based on ball possession, team networks and tactical actions classifications. Technical outcomes were mainly based on individual actions and skills characterized in quantity and success (i.e., successful passes, dribbles and shots) (Coutinho et al., 2020, 2022a, 2022b, 2022c; Folgado et al., 2019; Olthof, Frencken & Lemmink, 2018; Ric et al., 2017). Data collection was based on GPS, LPM and optical-based tracking systems ranging from 5–18 Hz. The first studies were mainly based on 5 Hz GPS (SPI-Pro X II, GPS ports, Canberra, ACT, and Australia). However, the use of sampling frequency at 5 Hz must consider some limitations in the measurement of non-linear and high-intensity paths (Portas et al., 2010; Teixeira et al., 2021a). Authors should prioritise tracking devices with sampling frequencies above 10 Hz shape with an accelerometer (Gómez-Carmona et al., 2020; Rico-González et al., 2020). The latest GPS devices already recommend a sampling frequency of 10–18 Hz, specifically 10 Hz GPS units (WIMU PRO; RealTrack Systems, Almeria, Spain) (Canton et al., 2021; Machado et al., 2022), 10 Hz GPS (S5; Catapult Innovations, Melbourne, Australia) (Folgado et al., 2019; Nieto et al., 2022), 10 Hz GPS (Polar Team Pro, Kempele, Finland) (Praça et al., 2021), 15 Hz GPS (SPIPRO; GPSports, Canberra, ACT, Australia) (Coutinho et al., 2017; Praça et al., 2016; Ric et al., 2016a) and 18.18 Hz GPS (GPEXE GK; Exelio SRL, Udine, Italy) (Jara et al., 2019). LPM devices and semiautomatic video tracking system used in the surveyed studies were the Prozone® (ProZone Holdings Ltd, Leeds, UK) (Folgado et al., 2015) and the Inmotio Object Tracking® (BV., Amsterdam, The Netherlands) (Olthof, Frencken & Lemmink, 2018). The integration of the different tracking systems can be further explored from an integrative perspective (Buchheit et al., 2014; Linke, Link & Lames, 2018). Also, the relationship between objective (i.e., tracking systems) and subjective (i.e., observational/notational analyses) measures should be explored in future integrative approaches, in order to make the integration of technical and tactical factors more effective (Teixeira et al., 2021b). Indeed, the non-positional data can act as an added value to the positional raw data by making the information gathered from the tracking systems more feasible and comprehensive (Praça et al., 2022; Teixeira et al., 2021b).

Likewise, internal training load was collected by heart-rate-based measures and perceived exertion. Although limitations have been reported in some studies, the perceived exertion and heart-rate maintains its feasibility in elite and sub-elite football settings, it is cost-effective and straightforward to employ (Achten & Jeukendrup, 2003; Teixeira et al., 2022a; Teixeira et al., 2021b). A study used portable optical timing system to measure neuromuscular performance (i.e., countermovement jump, CMJ). Field-based tests in football become more effective when continuous control (i.e. monitoring) is integrated into the assessment (Clemente et al., 2022; Gómez-Carmona et al., 2020).

Seven studies conducted null hypothesis statistical test (NHST) and five magnitude-based inferences (MBI). Some authors used both types of analyses, besides statistical analyses such as coefficient of variation (CV) (n = 7), effect sizes (ES) and smallest worthwhile changes (SWC) (Bernards et al., 2017; Flanagan, 2013). The application of the two statistical procedures (NHST and MBI) makes it more difficult to compare results of availed studies, and future pieces of research should further analyse the potential overvaluation and bias of the study findings (Foster, Rodriguez-Marroyo & de Koning, 2017; Welsh & Knight, 2015).

The current systematic review has limitations that should be considered. Firstly, the interpretation of the studies was only qualitative, not have been done a meta-analysis. Secondly, only studies that integrated physical and tactical measures were retained for further analysis, instead of just one dimension. For this reason, other details on physical and behavioural data may be absent (Duarte et al., 2013b; Folgado et al., 2014a; Vilar et al., 2014). Although this was made clear in the inclusion and exclusion criteria, future systematic reviews should clarify which studies used both one-dimensional and integrative approach. There are topics that can be explored in the future: (i) the development of user-friendly interfaces to depict positional data, because MATLAB® routines requiring extensive training in coding and programming are used to process and display data (Math-Works, Inc., Natick, MA, USA); (ii) developing tracking and wearables devices enabling real-time feedback to increase the practical applicability and decision making of football players and coaching staff; (iii) applying advanced data analytics and big data-based procedures using artificial intelligence, machine learning and deep learning to compute automatically physical and positional data; (iv) manipulating task constraints in MSG, LSG and different game-formats can still be better exploited; (v) woman’s football is still not analysed using physical and tactical integration.

Conclusions

Based on this systematic review, physical and tactical factors can be integrated by positional data using player’s movement speed, spatial movement (and their variability, regularity or predictability), complex indexes, playing areas, intra-team and inter-team synchronization dyads. Futures research should consider applying positional data in women’s football and explore the representativeness of the MSG and LSG in youth training settings. Although positional data is being extensively applied in semi-professional and professional football, user-friendly and real-time interfaces streaming physical and tactical outcomes should be consider to enable the widespread of this technology to all.

Supplemental Information

Supplemental Information 1 PRISMA checklist.

Click here for additional data file.

Supplemental Information 2 INPLASY registration of the protocol.

Click here for additional data file.

Supplemental Information 3 Rationale for conducting the systematic review / meta-analysis.

Click here for additional data file.

Additional Information and Declarations

Competing Interests

Author Contributions

Data Availability

Tiago M. Barbosa is an Academic Editor for PeerJ.

José Eduardo Teixeira conceived and designed the experiments, performed the experiments, analyzed the data, prepared figures and/or tables, authored or reviewed drafts of the article, and approved the final draft.

Pedro Forte conceived and designed the experiments, performed the experiments, authored or reviewed drafts of the article, and approved the final draft.

Ricardo Ferraz performed the experiments, authored or reviewed drafts of the article, and approved the final draft.

Luís Branquinho analyzed the data, authored or reviewed drafts of the article, and approved the final draft.

António José Silva conceived and designed the experiments, prepared figures and/or tables, authored or reviewed drafts of the article, and approved the final draft.

António Miguel Monteiro conceived and designed the experiments, prepared figures and/or tables, authored or reviewed drafts of the article, and approved the final draft.

Tiago M. Barbosa conceived and designed the experiments, authored or reviewed drafts of the article, and approved the final draft.

The following information was supplied regarding data availability:

This is a systematic review. All data is provided in the main text, notably in tables 1 to 6 all individual data from each paper retained for review is reported in a comprehensive fashion way.

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
