# Peer review of "Integrating physical and tactical factors in football using positional data: a systematic review"

_PeerJ, doi:10.7717/peerj.14381_

## Round 0.1 · original submission · Minor Revisions

Dear Authors

Your submission has been reviewed by two experts in the field of study. The comments of the reviewers are included at the bottom of this letter. We invite you to submit a revised version of the manuscript that addresses the points raised by the reviewers.

We look forward to receiving your revised manuscript.

Best regards

Yung-Sheng Chen, PhD
Academic Editor

Reviewer 1 ·

Basic reporting

Integrating physical, physiological and tactical factors in football using positional data: A Systematic Review.

Thank you for the opportunity to review this systematic review. I welcome studies that introduce novelty and applicability the important of Positional data have been used to capture physical, physiological and tactical factors in football. In fact, I am open to be persuaded to deep understand this relationship.
Hence, I am some sympathy with the author's intentions. In addition, the authors provide a decent description of the systematic review process. The topic represents contemporary interest, and the scope of the work is appropriate for PEER J. I think it is very important to conduct systematic review like this one, because in many occasions the different term used in different countries do that scientist perform different studies with different relevant mistakes. ITherefore, it is necessary to bring to light work whose aim is to review what has been published previously on this topic in a rigorous way. This study is particularly important because, as the authors indicate, the publications on this topic are increasingly, so it is necessary to group all these papers to have a better understanding of the results obtained previously.
From my point of view, the main strength of the manuscript is the methodology. The authors have followed properly the PRISMA guidelines. The systematic review has been included in the IMPLASY register. Apart from this general commentary about the manuscript, more details of some parts of the manuscript (strengths, weakness and questions) are found hereafter.

Experimental design

From my point of view, the main strength of the manuscript is the methodology. In this sense, the experimental design is correct.

Validity of the findings

The finding truly are relevants and support the previous research about the topic.

Additional comments

- For future investigations should be indicated at the end of the abstract as well as in the conclusions section some directions about the results in relation to soccer players.

Introduction
It is well written and structured. It is a good starting point to place the reader. However, it would be helpful to outline why your revision is important and clearly differentiated of other works. Moreover, it would be interesting if the bibliography was updated. Are you sure that the literature is enough?

Methodology
As it has been mentioned before, this section is the strongest part of the manuscript. The PRISMA methodology is clear
Why you use for quality assessment PEDro? I have not clear is you add crossover papers, parallel groups…. Therefore, in this case you need use other quality assessment as Rob 2 or ROBINS. But I have not clear because the more information you need add in PICOS? I Truly think that the information of PICOS is better for the reader in a table .
The gathered records were independently screened by title and abstract by two authors (JET and P). Additionally, they reviewed the full version of the included papers in detail to identify articles that met the selection criteria. An additional search of the reference lists of the included records was conducted to retrieve additional relevant studies.??

Discussion
- More information about the strong and limitations should be included.
- In table 4 and 5, less information due appear. Please reduce the information
- Check the note under the table.
Conclusions
The conclusions respond to the objectives of the systematic review but is too short. In fact, prospective about the RSMA should be included.

Reviewer 2 ·

Basic reporting

Introduction
L57: the amount of published research, much of them cited by the authors, contradicts this assumption. Indeed, much research has been published recently on using positional data for tactical purposes.
L71: the absence of literature on this integrative view is insufficient to justify a new study. I’d recommend that the authors expand on the current study's relevance. Also, it seems necessary to justify conducting a systematic review to answer the research problem instead of an experimental study.

Methods
L95: it is not clear how spatiotemporal data can be used to gather physiological information. Even if some of the available GPS devices allow integration with heart rate monitors, the physiological demands are not measured through GPS devices. I strongly recommend the authors review it.
The authors assumed PRISMA as the method. However, there are many missing topics in the methods. Therefore, it might be assumed that PRISMA was not fully adopted – or it was adapted. Please consider reviewing all the items from PRISMA or choosing a different method for reporting systematic reviews that better suits the current study.
L123: please indicate the items for quality assessment.
Please consider reviewing the items for quality assessment if they are entirely based on the reported studies. In some cases, observational studies in official matches simply cannot accomplish some criteria because it seems impossible to manipulate some things (for example, it is impossible to double-blind a data collection within an official match). Those scales were not meant to be applied in observational studies with official matches.

Results
Table 4: many of the reported dependent variables are impossible to be collected through spatiotemporal analysis (which is the aim of the current study). I recommend that the authors include only the data regarding the study’s aim.
Table 5:
Praça et al. (2016) did not analyze LSG (11-a-side). Please check it.
Finally, splitting the articles into those focused on training activities (such as SSGs) and actual match-play activities seems relevant. When analyzing the results, will facilitate the comparison between studies.

Discussion
Did the authors of the selected studies perform multidimensional analysis (for example, multivariate analysis) or just consider a multidimensional model to explain the performance (with many dependent variables)? This seems an exciting point. However, according to the results, previous studies did not fully address the problem of soccer being a multifactorial sports outcome.
L235: Praça et al. 2016 did not investigate LSG.
L250…: this is a result, not a discussion. What are the implications of adopting different frequencies? What does the literature say? Here the authors only pointed out what they achieved, but no explanations were provided.

Experimental design

See the previous topic.

Validity of the findings

See the previous topic.

Additional comments

See the previous topic.

---

## Round 0.2 · Minor Revisions

Dear Authors

There are a number of typos and grammatical errors in the manuscript. It should be edited before acceptance.

For example:
- 'researches' in the abstract should be 'research'
- 'the players collect ecological information’s' (should be 'information')
'Currently, a growing reviews and meta-analysis have been published' (should be '...a growing number of reviews and metaanalyses...')
- 'The literature search was accessed...' (should be 'performed'
etc.

Best Regards

Yung-Sheng Chen, PhD
Academic Editor


Reviewer 1 ·

Basic reporting

After the changes made, I consider that the work is ready for publication.

Great job

Experimental design

All is correct

Validity of the findings

The use of systematic reviews is always positive for the references extension This study provides knowledge about the use of gps in soccer and

Reviewer 2 ·

Basic reporting

Nothing else to address.

Experimental design

Nothing else to address.

Validity of the findings

Nothing else to address.

Additional comments

Nothing else to address.

---

## Round 0.3 · accepted · Accept

Dear Authors

I'm pleased to inform you that your manuscript is accepted for publication.
Congratulations.

Best Regards

Yung-Sheng Chen, Ph.D.
Academic Editor